# Japanese Translation and Validation of Genomic Knowledge Measure in the International Genetics Literacy and Attitudes Survey (iGLAS-GK)

**DOI:** 10.3390/genes14040814

**Published:** 2023-03-28

**Authors:** Akiko Yoshida, Tomoharu Tokutomi, Akimune Fukushima, Robert Chapman, Fatos Selita, Yulia Kovas, Makoto Sasaki

**Affiliations:** 1Iwate Tohoku Medical Megabank Organization, Iwate Medical University, Shiwa 020-3694, Japan; akikoyos@iwate-med.ac.jp (A.Y.);; 2Department of Clinical Genetics, School of Medicine, Iwate Medical University, Morioka 020-8505, Japan; 3Department of Psychology, Goldsmiths, University of London, 8 Lewisham Way, London SE14 6NW, UK

**Keywords:** genetics, genomic knowledge, international genetics literacy and attitudes survey, Japanese

## Abstract

Knowledge of genetics is essential for understanding the results of genetic testing and its implications. Recent advances in genomic research have allowed us to predict the risk of onset of common diseases based on individual genomic information. It is anticipated that more people will receive such estimates of risks based on their genomic data. However, currently, there is no measure for genetic knowledge that includes post-genome sequencing advancements in Japan. In this study, we translated the genomic knowledge measure in the International Genetics Literacy and Attitudes Survey (iGLAS-GK) into Japanese and validated it in a general Japanese adult population (*n* = 463). The mean score was 8.41 (SD 2.56, range 3–17). The skewness and kurtosis were 0.534 and 0.088, respectively, and the distribution showed a slightly positive skewness. Exploratory factor analysis proposed a six-factor model. Results for 16 of the 20 items of the Japanese version of the iGLAS-GK were comparable to those from previous studies in other populations. These results indicate that the Japanese version is reliable and can be used to measure the genomic knowledge of adults in the general population, and this version of the knowledge measure maintains the multidimensional structure for assessing genomic knowledge.

## 1. Introduction

It has been demonstrated that genomic information has potential for application in the primary care of common diseases, such as heart disease and diabetes [1,2]. This has been enabled by the development of polygenic risk scores that predict the risk of a disease based on multiple disease-associated genetic markers [3,4,5,6,7]. With increasing availability of genetic information on common diseases in primary care, many patients will be confronted with their own genetic information when making decisions regarding their healthcare. Therefore, understanding the level of genomic knowledge of the public is essential for supporting their decisions and improving adherence to preventive interventions [8].

To date, several instruments have been developed to measure genetic knowledge [9,10,11,12,13,14,15]. However, most of these instruments rely on a Mendelian view of the genetic contribution to disease phenotypes and do not include items about polygenicity. In this milieu, the International Genetics Literacy and Attitudes Survey (iGLAS)-GK, a measure of knowledge of genomics, includes items on topics such as polygenicity, heritability, and epigenomics [16]. The iGLAS-GK—a part of the iGLAS—was developed in the UK; it is designed to capture the perceptions and knowledge of the general population, and legal and educational professionals regarding genetics. The iGLAS consists of items on genomic knowledge and attitudes toward genetics-related technologies and legislations. It also serves as an educational resource as, at the end of testing, it provides correct answers, explanations, and sources of information pertaining to the questions. The iGLAS is currently available in eight languages (Albanian, English, French, Italian, Persian, Romanian, Russian, and Spanish) [17].

In Japan, there is no such knowledge measure that includes the concepts of genomics and population genetics. Therefore, in this study, we translated the contents in the iGLAS-GK into Japanese and validated it in a general Japanese adult population with the aim to develop a genomic knowledge measure that includes post-genome sequencing advancements.

## 2. Materials and Methods

### 2.1. Instruments

In this study, iGLAS-GK version 3 was used. The previous version of iGLAS-GK consisted of 18 items, whose reliability and validity have been confirmed [16,18]. In iGLAS-GK version 3, 2 of the 18 items were excluded, and 4 new items were added; thus, there were 20 items in total. We developed a Japanese version of iGLAS-GK version 3 and confirmed the equivalence between the original and Japanese versions through four steps: (i) initial translation from English to Japanese, (ii) back-translation and feedback by a translator who is a native English speaker and non-expert in genetics, (iii) discussion of the pre-final version with six Japanese experts (2 clinical geneticists, 3 genetic counselors, and 1 genome medical research coordinator) and four students of genetics, and (iv) confirmation of the final version by the authors of the iGLAS.

### 2.2. Participants

Participants were recruited from the general public who had registered as Internet survey monitors with iBridge Corporation, an online survey company. Emails were sent to monitors for recruitment. Six hundred respondents were enrolled in the item analysis (Study 1) and 100 were enrolled in the test–retest analysis (Study 2). Both study groups were composed of respondents with the same number of male and female participants aged 20–69 years in each of the five 10-year age groups. Studies 1 and 2 were conducted on the website of the survey company. None of the items had “I don’t know” options so that all items required a response, but the participants were allowed to discontinue at any time. Data from the respondents who did not respond to all of the statements were not recorded. Study 1 was conducted on 22 February 2022. For Study 2, the first test was performed from 25 February to 28 February 2022, and the second test was conducted from 28 April to 11 May 2022—8 weeks from the first test. The time interval was decided based on a previous study of the iGLAS [16].

For item analysis (Study 1), the desired number of participants was >384. This figure was calculated using the sample size calculation for a study of a dichotomous variable (correct or incorrect answer) [19]. The conditions for the calculation were as follows: expected proportion of correct answers = 0.50; 95% confidence interval (95% CI) = 0.10. For the test–retest analysis (Study 2), the required number of participants was >19 when the correlation coefficient between the first and second tests was assumed to be 0.6 with a significance level of 0.05 and a power of 0.80. We assumed a correlation coefficient based on previous studies on the genetic knowledge measure [16,20].

To ensure sufficient power, we recruited more participants than the calculated number, accounting for foreseeable data collection interferences, such as participants intentionally or unintentionally not responding to some items, answering without reading the text, answering randomly, or dropping out [21,22]. Based on experience, we estimated that approximately 30% of participants would submit unreliable data, such as responding without reading the statements, and approximately 30% would drop out of the second test of Study 2.

### 2.3. Measure

The measure consisted of 20 items from the iGLAS-GK (see Table 1 for the list of items). Additionally, we included one statement about educational background and two statements to identify unreliable data: the instructional manipulation check and seriousness check (see details below). The 20 items of the iGLAS-GK had either four or two response options. Each item was designed to provide a single answer. One point was assigned for a correct answer. The maximum score was 20 when the respondent answered all questions correctly. Data on sex, age, and time taken to answer were obtained from the survey company. For the purposes of this study, the answers and explanations to the statements were not provided to the participants.

### 2.4. Identification of Unreliable Responses

Data from the following participants were excluded from the analysis: participants who took <40 s to answer all statements, those who chose an option not indicated by the instructional manipulation check, and those who answered “did not respond seriously” to the seriousness check. The cut-off time, <40 s, was determined as the average time taken by five graduate students in the Department of Clinical Genetics at Iwate Medical University to respond to all statements on their smartphone without reading the text. The wording of the instructional manipulation check was: “The following options are presented to confirm that you read the statements. Please choose 40 here”. Participants were able to choose one of four answers: “10”, “20”, “30”, and “40”. The wording of the seriousness check was: “There is a problem with online-based surveys where some people give answers without reading the statements. It would be very helpful for data analysis if you could tell us whether you have taken part seriously”. Participants were able to choose one of two answers: “I have taken part seriously” and “I have not taken part seriously”. The instruction manipulation check was presented between Q19 and Q20 and the seriousness check was presented at the end of the statements.

### 2.5. Statistical Analysis

#### 2.5.1. Item Analysis

We confirmed item difficulty by estimating the percentage of correct answers for each item. Furthermore, we estimated the item discrimination index for each item by performing the good–poor analysis. In the good–poor analysis, we compared the correct answers of the following two groups: (1) respondents whose scores were in the first quartile of the score distribution or below and (2) respondents with scores in the third quartile or above.

#### 2.5.2. Ceiling and Floor Effects

Ceiling and floor effects were also examined. These effects were considered to be present when the sum of the mean score and SD was >20 points (ceiling effect) or when the value obtained by subtracting the SD from the mean was <0 (floor effect). Skewness and kurtosis were calculated to confirm score distribution.

#### 2.5.3. Test–Retest

To verify the stability of the measure, Spearman’s rank correlation coefficient between the first and second tests was calculated in Study 2.

#### 2.5.4. Confirmation of Factor Structure

To examine the factor structure of the 20-item measure, a parallel factor analysis was performed to identify the number of factors. In addition, factor analysis using the maximum likelihood method with Oblimin rotation was carried out assuming a six-factor model to confirm factor structure and the inter-factor correlations.

#### 2.5.5. Significance Level and Analysis Tools

For all the statistical analyses indicated above, the cut-off for statistical significance was set at *p* < 0.05. Statistical analyses were performed using R (version 4.0.3). The semTools package (version 0.5-5) was used to calculate skewness and kurtosis. The psych package (version 2.0.9) was used for parallel factor analysis.

### 2.6. Ethical Considerations

Information on the purpose of this study and consent items was displayed before proceeding to the statement pages of the items. The participants had to click the “accept” button to provide consent and proceed. This study was conducted in accordance with the tenets of the Declaration of Helsinki and the protocol was approved by the Ethics Committee of the Medical Faculty of Iwate Medical University (no. MH2021-193 for Study 1; no. MH2021-194 for Study 2).

## 3. Results

### 3.1. Content Validity

The Japanese version of contents in the iGLAS-GK was developed by translating, back-translating, and receiving feedback from the panel of expert and non-expert contributors, as well as authors of iGLAS to ensure equivalence with the original.

### 3.2. Participants

Six hundred participants responded to the statements in Study 1. In Study 2, wherein the survey was conducted twice, 100 participants completed the responses in the first survey, and of these, 89 completed the responses in the second survey. After excluding unreliable data, valid responses were obtained from 463 participants (77.2%) for Study 1 and from 48 participants (53.9%) for Study 2. The demographic characteristics of the participants are presented in Table 2.

In Study 1, there were 234 (50.5%) male participants and 229 (49.5%) female participants, of mean age 45.6 and 46.4 years, respectively (range: 20–69 years for participants of both sexes). The education levels were as follows: junior high school, 8 (1.7%); senior high school or its equivalent, 121 (26.1%); vocational school or technical college, 93 (20.1%); bachelor’s degree, 212 (45.8%); master’s degree 20 (4.3%); and doctoral degree, 9 (1.9%). In Study 2, there were 21 (43.8%) male and 27 (56.3%) female participants, of mean age 47.4 and 47.6 years, respectively (range: male participants, 22–66 years; female participants, 27–65 years).

### 3.3. Item Analysis

The item difficulty and item discrimination for each item in Study 1 are presented in Table 1. Item difficulty is indicated by the proportion of participants answering an item correctly. Correct responses ranged from 14.3% to 68.5% (mean 33.7%, SD 15.1) for items with four choices and from 47.1% to 87.7% (mean 61.5%, SD 15.4) for items with two choices. The overall mean of the 20 items was 42.0%, and the SD was 19.7. In all, 16 of the 20 items had a difficulty level between 20% and 80%. Three items, namely Q6, Q10, and Q20, were very difficult, with a correct response rate of less than 20%, whereas Q17 was very easy, with a correct response rate of 87.7%.

The good–poor analysis was performed and item–total correlations were calculated to validate item discrimination. In the good–poor analysis, the difference in item difficulty between the upper- and lower-scoring groups ranged from 2.4% to 47.6%. Except for one item (Q6), differences between the two groups for all items were significant.

In the item analysis, 16 items met the following criteria: item difficulty ranging from 20% to 80% and significant differences found in the upper and lower group comparisons. Q10, Q17, and Q20 met only the discrimination criteria, whereas Q6 did not meet either of the criteria.

### 3.4. Number of Factors and Inter-Factor Correlations

An exploratory factor analysis was performed to examine the number of factors and correlations between the factors. The parallel analysis suggested a six-factor model. Factor analysis was performed based on the six-factor model. Factor loading and communality are presented in Table 1. Thirteen items showed factor loading with an absolute value of 0.4 or higher on a factor. Twelve of the thirteen items had positive factor loading values, and one item (Q20) had a negative value. The factors were named as follows: First factor (Q7 and Q11): “Multifactorial diseases”; Second factor (Q6, Q13, and Q20): “Variability”; Third factor (Q18 and Q19): “Genetic effects on traits”; Fourth factor (Q2, Q10, Q14, and Q16): “Genome”; Fifth factor (Q5): “Gene function”; Sixth factor (Q17): “Genetic determinism”. Correlations between any two factors were less than 0.2, as the absolute value and the factor axes were almost orthogonal. The cumulative contribution ratio was 46.3%.

### 3.5. Summary Statistics of the 20 Items

Figure 1 shows the distribution of total scores for the 20 items. The mean score was 8.41 (SD 2.56, range 3–17), showing no ceiling or floor effects. The distribution was slightly positive, with a skewness of 0.534 and kurtosis of 0.088. There were no significant differences in scores according to sex, age, or educational background (Appendix A).

### 3.6. Test–Retest Reliability

The test–retest method was used to examine the stability of the measure (Study 2). The mean scores for the first and second tests were 8.15 (SD = 2.51) and 9.00 (SD = 2.74), respectively. A significant test–retest correlation was confirmed using Spearman’s rank correlation coefficient (0.55; *p* < 0.001, 95% CI [0.31–0.72]).

## 4. Discussion

In this study, we aimed to evaluate the Japanese version of iGLAS-GK, a genomic knowledge measure for the post-genome era. The score distribution of the 20 items of the iGLAS-GK showed no significant distortion. Of the 20 items, 19 had adequate item difficulty and/or discrimination, and the test–retest study showed sufficient stability. These results demonstrate that the Japanese version of the iGLAS-GK is an acceptable research instrument that can be used for the general Japanese adult population.

Item difficulty and discrimination depend on the examinee sample [23]. Therefore, proper sampling of the target population is necessary for item analysis. In this study, we targeted the general adult population of Japanese male and female individuals. The proportion of male to female participants was nearly equal (Table 2). In addition, the responses showed no significant differences across the five age groups. The participants comprised five approximately equal-size age groups for each decade, with ages ranging from the 20s to the 60s, even after excluding participants who submitted unreliable data.

Our sample was comparable with that of the 2020 national census of Japan [24], in which the percentage of participants of each age group, from 20 to 69 years, ranged from 15% to 24%, and the male-to-female ratio was 1.0 for all age groups. However, in the national census, 78% had completed high school or higher education, which is lower than the proportion in our study. The overall higher level of education in the present study could be attributed to the requirement for participants to complete all items. It is likely that participants with a lower level of education were overall less confident in their ability to answer all items and, therefore, were less likely to proceed to the end of the test.

Overall, the participants of this study constituted an appropriate sample population for validating the Japanese version of iGLAS-GK, which was originally developed for general adult populations.

Regarding item difficulty and discrimination, sixteen of the 20 items had correct response rates between 20% and 80%. The average of correct responses for the 20 items was 42.0%. The original validation study of the iGLAS-GK showed an average correct responses level of 66% [18]. In other genetic knowledge measures, the correct response rates varied from 43% to 73.3% [14,20,25]. Some measures can identify individuals with low scores, whereas others discriminate individuals with high and low scores.

The optimum item difficulty to discriminate respondents with high and low scores is the median between 100%, where all respondents choose the correct answer, and chance, where respondents choose the correct answer by guessing [26]. Based on this definition, the 62.5% and 75.0% accuracy rates represent the optimum item difficulties for items with four and two response options, respectively. In the iGLAS-GK, the correct responses of the 14 items with four response options ranged from 14.3% to 68.5%. The responses for the six items with two response options had accuracy rates ranging from 47.1% to 87.7%. Overall, the response rates for 18 items, excluding Q5 and Q17, indicated greater difficulty than optimum.

The distribution of the total score of the 20 items was slightly positively skewed, with a skewness of 0.534, and a tail tending toward the high-scoring side (Figure 1). However, the mean score for the 20 items was 8.41, with an SD of 2.56, indicating no ceiling or floor effect. These results show that the Japanese iGLAS-GK version is suitably difficult for adults in the general population.

Consistent with the appropriate item difficulty, 19 of the 20 items, excluding Q6, showed significant differences between the lower and upper score groups in the good–poor analysis. These results indicate that the Japanese version of iGLAS-GK can discriminate individuals based on their scores, albeit with one item that have a weak discriminative power.

In a previous study, factor analysis suggested that the iGLAS-GK did not support a single-factor model, indicating that genomic knowledge is likely to represent a diverse knowledge base for those who have not specifically studied genetics [16]. Six factors were proposed by factor analysis for the 20 items of the iGLAS-GK in this study, cumulatively explaining 46.3% of the variance in genomic knowledge. Thirteen items showed factor loading with an absolute value of 0.4 or higher on a factor. The dimensions of the genomic knowledge represented by the six factors are “Multifactorial diseases”, “Variability”, “Genetic effects on traits”, “Genome”, “Gene function”, and “Genetic determinism”. These item clusters cover the key concepts of genome structure and function, including epigenetic processes and gene–environment interplay.

The second factor “Variability” showed positive factor loadings for Q6 (inter-individual genetic differences) and Q13 (within individual inter-cell sequence similarity-epigenetic difference); and negative loading for Q20 (understanding heritability). Heritability is a value that indicates how much genetic information influences the variance of a phenotype in a population. To answer Q20 correctly, it is necessary to understand not only the diversity of genomes and gene expression at the individual level but also the variance of phenotypes at the population level and the influence of genetic and environmental factors. Research shows that heritability remains a very difficult and often misunderstood concept [27,28].

Among the seven items with lower values of the factor loading, Q3, Q4, and Q8 assessed understanding of the genome structure; Q1 and Q12 measured understanding of the genome and its variation; and item 15 evaluated knowledge of genetic engineering (Q15). Item 9 that specifically related to epigenetics showed low accuracy rate and did not cluster with other items that also indirectly tapped into understanding of epigenetics (Q 13 and 17). This suggests overall poor understanding of epigenetic processes, including cell differentiation and environmental influences on gene expression. This knowledge is particularly important in disease etiology, and in disease prevention, maintenance, and treatment. However, epigenetics is a relatively new field compared with genetics, and it is expected that people will increasingly gain knowledge about it in the future. Overall, multiple factors which influence genetic literacy, including science education, media trends, and people’s interest in scientific and technological developments, might be intricately involved.

The six axes were almost orthogonal, indicating only a weak correlation between any two factors in the iGLAS-GK. These results support the notion that the measure taps into a wide range of genetics-related matters, from scientific basics to social applications, and that multiple factors underlie the genomic knowledge across different contexts. These results also show that there are more than six factors to explain the variance of genomic knowledge, indicating the complexity of its multifactorial structure. Increasing genetic literacy is likely to lead to a more homogeneous structure of knowledge, so that for participants who have a better understanding of the key genetic concepts, the concepts become organized into a coherent knowledge system, rather than a collection of facts.

Sixteen (Q1–Q14, Q17, and Q18) of the 20 items were included in the iGLAS-GK of previous studies that reported the reliability and validity of the measure [16,18]. Compared with those of the original validation study [18], out of the 16 items, the participants of this study had low accuracy rates for 14 and high accuracy rates for 2 items. This result demonstrates that the iGLAS-GK can indicate differences in genomic knowledge among different samples.

In the present study, three (Q6, Q10, and Q17) of the 16 items had accuracy rates outside the range of 20% to 80%. In contrast, in previous research, the accuracy rates of these items ranged from 45% to 63% [18]. Beyond differences in the participants’ education levels, a possible reason for the difference in the difficulty of these items may be the different approaches used to recruit participants. In the previous research [18], participants were recruited through social media platforms, such as Facebook, online science forums, and email, through a network community of educators. In the present study, participants were recruited through a monitor panel of an online research company. Therefore, the participants in this study may represent a broader population with a wider variety of levels of interest in genetics than those in the previous study. Different results may be obtained when iGLAS-GK version 3 is applied to a specific population that differs from the general population.

The iGLAS-GK was revised in consultation with genetics experts. Therefore, the 20 items of iGLAS-GK version 3 retain face validity.

Overall, given these results, and considering that the iGLAS-GK was revised in consultation with genetics experts and retains face validity, all 20 items can be included in the Japanese version of iGLAS-GK.

The limitation of the current study is that the item analysis method used was based on the classical test theory. According to the classical test theory, item difficulty and discrimination are relative values that depend on the properties of the sample group [23]. Therefore, it would be necessary to validate the measure in subgroups of the Japanese population, such as specific age or professional groups, or patients with a particular disease, if the iGLAS-GK should be used for those subgroups in the future.

## Figures and Tables

**Figure 1 genes-14-00814-f001:**
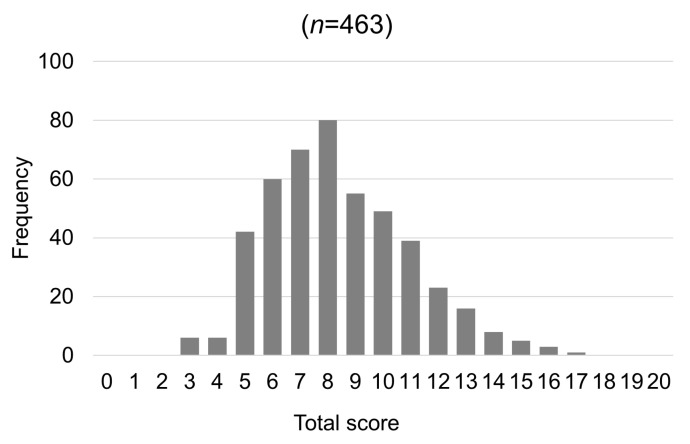
Distribution of the total scores.

**Table 1 genes-14-00814-t001:** Item analysis and factor analysis of Japanese version of iGLAS-GK.

Question		Item Difficulty (%)				Good–Poor Analysis			Factor Analysis ^b^						
		Total	Lower (*n* = 184)	Middle (*n* = 135)	Upper (*n* = 144)	Subtraction (Upper-Lower)	*p* Value ^a^	95% CI	Factor Loading						Communality
									F1	F2	F3	F4	F5	F6	
Q1 †	What is a genome?	36.1	21.2	38.5	52.8	31.6	<0.001 ***	[21.5, 41.6]	0.00	0.10	0.13	0.11	0.12	0.05	0.07
Q2 †	Which of the following 4 letter groups represent the base units of DNA?	42.5	21.2	43.7	68.8	47.6	<0.001 ***	[38.0, 57.2]	−0.05	0.00	−0.03	**0.61**	0.19	−0.02	0.45
Q3 †	How many copies of each gene do we have in each autosome cell?	21.6	11.4	19.3	36.8	25.4	<0.001 ***	[16.3, 34.5]	0.18	0.06	−0.01	0.36	−0.08	−0.08	0.15
Q4 †	People differ in the amount of DNA they share. How much of this differing DNA do siblings usually share?	44.7	28.8	46.7	63.2	34.4	<0.001 ***	[24.1, 44.6]	0.02	−0.36	0.00	0.26	0.03	0.03	0.20
Q5 †	What is the main function of all genes?	68.5	52.2	69.6	88.2	36.0	<0.001 ***	[27.1, 45.0]	−0.01	−0.02	0.01	0.02	**0.99**	−0.02	1.00
Q6 †	On average, how much of their total DNA is the same in two people selected at random?	14.3	13.6	13.3	16.0	2.4	0.544	[−5.4, 10.2]	−0.01	**1.00**	−0.01	0.00	−0.04	−0.04	1.00
Q7 ‡	Genetic contribution to the risk for developing Schizophrenia comes from:	57.2	46.7	55.6	72.2	25.5	<0.001 ***	[15.2, 35.8]	**0.77**	−0.08	−0.08	0.02	−0.04	0.03	0.63
Q8 †	In humans, DNA is packaged into how many pairs of chromosomes?	40.0	26.1	37.0	60.4	34.3	<0.001 ***	[24.1, 44.5]	−0.23	0.12	0.02	0.34	0.19	0.08	0.28
Q9 †	An epigenetic change is:	33.3	28.8	32.6	39.6	10.8	0.040 *	[0.5, 21.1]	−0.23	−0.05	−0.02	0.01	0.19	−0.21	0.13
Q10 †	Approximately how many genes does the human DNA code contain?	17.7	12.5	8.9	32.6	20.1	<0.001 ***	[11.1, 29.2]	0.07	0.22	−0.10	**0.40**	−0.09	0.11	0.23
Q11 ‡	Genetic contribution to the risk for developing Autism comes from:	56.4	46.2	52.6	72.9	26.7	<0.001 ***	[16.5, 36.9]	**1.00**	0.04	0.02	−0.01	0.02	−0.02	1.00
Q12 †	What are polymorphisms?	49.9	32.1	53.3	69.4	37.3	<0.001 ***	[27.3, 47.5]	0.06	0.03	0.24	0.23	0.03	0.32	0.23
Q13 †	The DNA sequence in two different cells, for example a neuron and a heart cell, of one person, is:	21.0	13.0	16.3	35.4	22.4	<0.001 ***	[13.2, 31.6]	0.14	**0.47**	0.15	0.05	0.12	0.06	0.27
Q14 †	Non-coding DNA describes DNA that:	36.5	14.7	42.2	59.0	44.3	<0.001 ***	[34.8, 53.9]	0.13	−0.11	0.10	**0.48**	0.19	0.05	0.34
Q15 ‡	Can dog breeding be considered a form of gene engineering?	49.2	32.6	47.4	72.2	39.6	<0.001 ***	[29.6, 49.6]	−0.04	−0.06	0.34	0.11	0.11	−0.22	0.23
Q16 †	Which of the mentioned below is a method for gene editing:	28.7	20.7	26.7	41.0	20.3	<0.001 ***	[10.4, 30.3]	−0.08	−0.05	0.02	**0.53**	−0.35	−0.07	0.35
Q17 ‡	Can we fully predict a person’s behaviour from examining their DNA sequence?	87.7	84.2	86.7	93.1	8.9	0.014 *	[2.1, 15.5]	−0.01	−0.02	0.00	−0.01	−0.01	**1.00**	1.00
Q18 ‡	At present in many countries, new born infants are tested for certain genetic traits.	47.1	31.0	48.1	66.7	35.7	<0.001 ***	[25.5, 45.9]	0.02	−0.02	**0.55**	−0.05	0.02	−0.16	0.34
Q19 ‡	Some of the genes that relate to dyslexia also relate to ADHD:	71.3	50.5	80.7	88.9	38.4	<0.001 ***	[29.5, 47.2]	−0.02	0.01	**1.00**	0.00	−0.01	0.03	1.00
Q20 †	If a report states ‘the heritability of insomnia is approximately 30%,’ what would that mean?	17.1	11.4	21.5	20.1	8.7	0.029 *	[0.7, 16.7]	0.08	**−0.47**	0.10	0.01	−0.28	−0.16	0.36
		Mean (SD)							Percent of variance (%)						
		42.0 (19.7)	29.9 (18.4)	42.0 (21.3)	57.5 (22.4)				9.0	8.4	7.9	7.3	7.2	6.5	
		Mean † (SD)							Cumulative variance (%)						
		33.7 (15.1)	22.0 (11.2)	33.5 (17.1)	48.8 (20.5)				9.0	17.4	25.3	32.6	39.8	46.3	
		Mean ‡ (SD)													
		61.5 (15.4)	48.6 (19.2)	61.9 (17.3)	77.7 (10.6)										

Note. Item difficulty indicates the correct answer rate. The lower, middle, and upper score groups included participants with a score ≤7, 8–9, and ≥10, respectively. (^a^) Hypothesis testing for differences in population proportions. (^b^) Factor analysis using the maximum likelihood method with Oblimin rotation. The values of factor loading ≤−0.4 or ≥0.4 are bolded. F1: Multifactorial diseases, F2: Variability, F3: Genetic effects on traits, F4: Genome, F5: Gene function, F6: Genetic determinism. * *p* < 0.05, *** *p* < 0.001. † Items with four options, ‡ Items with two options.

**Table 2 genes-14-00814-t002:** Demographic features of respondents, *n* (%).

Characteristic	Study 1 (*n* = 463)	Study 2 (*n* = 48)
Sex		
Male	234 (50.5)	21 (43.8)
Female	229 (49.5)	27 (56.3)
Age		
20–29	81 (17.5)	7 (14.6)
30–39	87 (18.8)	10 (20.8)
40–49	92 (19.9)	7 (14.6)
50–59	96 (20.7)	10 (20.8)
60–69	107 (23.1)	14 (29.2)
Average	46.0	47.5
Educational background		
Junior high school	8 (1.7)	2 (4.2)
Senior high school or its equivalent	121 (26.1)	12 (25.0)
Vocational school or technical college	93 (20.1)	6 (12.5)
Bachelor’s degree	212 (45.8)	25 (52.1)
Master’s degree	20 (4.3)	3 (6.3)
Doctoral degree	9 (1.9)	0 (0.0)
Occupation		
Company employee (full-time)	165 (35.6)	15 (31.3)
Homemaker	78 (16.8)	9 (18.8)
Unemployed	69 (14.9)	9 (18.8)
Part-time work	48 (10.4)	5 (10.4)
Company employee (temporary staff)	23 (5.0)	2 (4.2)
Self-employed	20 (4.3)	0 (0.0)
Student	14 (3.0)	1 (2.1)
Freelancer	11 (2.4)	2 (4.2)
Manager/Executive	10 (2.2)	0 (0.0)
Physician/medical personnel	8 (1.7)	2 (4.2)
Public employee (excluding faculty/staff)	8 (1.7)	1 (2.1)
Other	9 (1.9)	2 (4.2)

## Data Availability

The datasets used and analyzed during the current study are available from the corresponding author on reasonable request.

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
