# Peer review of "Japanese Translation and Validation of Genomic Knowledge Measure in the International Genetics Literacy and Attitudes Survey (iGLAS-GK)"

_genes, 2023, doi:10.3390/genes14040814_

Round 1

Reviewer 1 Report

 will be pasting in my opinion on this paper, which depends heavily iGLAS-GK on a scale of international genomic literacy. The first assumption in the paper is that genetics can impact disease and it can but most chronic disease is genetically heterogeneous and the result of poor health choices in Japan and the world. For example, large amounts, amounts of smoked food, lead to stomach cancer, and other cancers I don't know that any studies of the genetics of that taste or salty, smoke taste, but despite the education without a neuropsychiatric intervention, I don't believe the Japanese can impact this disease. Most of these  chronic lethal diseases impact our descendants through epigenetics and behavioral choices. I don't believe the questionnaire being used really tells people that they smoke. It will affect their generations. It will not change their eyecolor. Therefore, epigenetic impact is that if patients use drugs eat poorly, it affects their kids lives for generations, depending on the situation, 1 to 3 generations of epigenetic's .    In a patient's lifetime epi-genetics are more significant than their genetic make up, which may be able to be paid forward to their descendants, and backward, we understand their ancestors and diseases that information is important, but as eugenics has proven, nobody can predict most disease by genetics alone most of the diseases in our society. So I believe the premises falls and requires these caveats!    Diseases are heterogeneous, and the biggest impact is on behavior and utilization of Neuropsychiatric techniques to change behavior so that epigenetic transmission of altered gene expression does not occur. The literacy test may need to be re-crafted to meet the needs of new research that should refocus, American, and world genetics in the area of compliance to health recommendations an implication of epigenetic damage.    Other questions that need to be asked either in another scale or why is public health deteriorating question? Americans gets fatter and fatter each generation ,  has become increasingly addicted and cognitively reduced to some degree. Even if the author doesn't agree with these promises again, caveats on, it's likely the scale used in this paper is severely insufficient in the area that's most important to healthcare our own behavioral choices and literacy is most needed in the area of epigenetic impact on your descendants.    More significant for the paper is you have a questionnaire that I had to look, for there should be an exact table of of the questionnaire, with each question numbered, I had to find a table of questions and link it up to their table of data. I think that's very difficult. The questions are not that long, and they should be putting short form in the paper next to the data table and the p values, etc  it's unreasonable to expect anyone to follow each question and it's separate analysis, especially when the overall global questionnaire does not appear to have the real validity and reliability and utility health utility data       then a table of the findings where the questions are referred to so the findings can be easily located. I am sympathetic to testing patient literacy, since there is very little literacy And I hope the papers published so that it would encourage health policy to check the literacy of all the major illnesses that cost the western world and industrial countries, productivity   Am I reading the data correct that the atheists have better genetic literacy, then religious people?  Is genetic Literacy than Christians I don't know if this is statistically significant, but it is significant for society in general, ? There's a huge cultural divide between Christians and non-Christians in America, and they need to realize even those of us who are sympathetic to Christianity must note the scientific study is diminished, but I need to know if it's statistically significant but at least a brief discussion of the possibility that the religious or not keeping up scientifically this is true, and Manny countries.

Reviewer 2 Report

This is a very straightforward study that is suitable for publication, if the editor finds it worthy of interest. I make just one comment.

The authors use the item-test correlation. This is an approximation of the item’s loading on the general factor measured by all items (McDonald, 1999). A problem is that the questionnaire measures six factors that are not strongly correlated, suggesting that any general factor is very weak in this relatively brief questionnaire. The item-test correlation is therefore not a good measure of the item’s discriminating power. A better measure would be an item’s loading on the one (of six) factors that it measures.

McDonald, R.P. (1999). Test theory: A unified treatment. Erlabum.
